# Characterization of Al-12Si Thin-Wall Properties Fabricated with Laser Direct Energy Deposition

Raihan Rumman [1], Mallaiah Manjaiah [2,3,*], Stéphane Touzé [2], Ruby Alice Sims [1], Jean-Yves Hascoët [2] and Jamie Scott Quinton [1,4]

1    Institute for NanoScale Science and Technology, Flinders Microscopy and Microanalysis, Flinders University, Adelaide 5042, Australia; jamie.quinton@flinders.edu.au (J.S.Q.)
2    Ecole Centrale de Nantes, CNRS, GeM, UMR 6183, F-44000 Nantes, France; jean-yves.hascoet@ec-nantes.fr (J.-Y.H.)
3    Department of Mechanical Engineering, National Institute of Technology Warangal, Warangal 506004, India
4    School of Natural Sciences, Massey University, Palmerston North 4442, New Zealand
*    Correspondence: manjaiah.m@nitw.ac.in; Tel.: +91-9740847669

**Abstract:** Additive manufacturing is an emerging process that is used to manufacture industrial parts layer by layer and can produce a wide range of geometries for various applications. AM parts are adopted for aerospace, automobiles, antennas, gyroscopes, and waveguides in electronics. However, there are several challenges existing in manufacturing Al components using the AM process, and their mechanical and microstructural properties are not yet fully validated. In the present study, a gas-atomised powder of a eutectic Al-12Si alloy was used as feedstock for the Laser Direct Energy Deposition (LDED) process. A SEM analysis of Al-12Si powder used for processing illustrated that particles possess appropriate morphology for LDED. A numerical control system was used to actuate the deposition head towards printing positions. The deposited samples revealed the presence of Al-rich and Al-Si eutectic regions. The porosity content in the samples was found to be around 2.6%. Surface profile roughness measurements and a microstructural analysis of the samples were also performed to assess the fabricated sample in terms of the roughness, porosity, and distribution of Al and Al/Si eutectic phases. The tensile properties of fabricated thin walls were better compared to casted Al alloys due to the uniform distribution of Si in each layer. Micro-hardness tests on the deposited samples showed a hardness of 95 HV, which is equivalent to casted and powder bed fusion melting samples. The gas atomised Al-12Si powders are highly reflective to a laser and also quick oxidation takes place, which causes defects, porosity, and the balling effect during fabrication. The results can be used as a base guide for the further fabrication of aerospace component design with high structural integrity.

**Keywords:** Laser Direct Energy Deposition (LDED); additive manufacturing; Al-Si eutectic alloys; microstructure; surface profile roughness; hardness

## 1. Introduction

Al-Si alloys have been widely used in automotive, aerospace, military, and domestic applications because of their high wear and corrosion resistance, low thermal expansion coefficient, and high strength [1]. Depending on the Si content within the Al-Si alloy composition, they can be classified as hypoeutectic (<11 wt.%), eutectic (11–13 wt.%), and hypereutectic alloys (>13 wt.%) [2]. The Si content thus plays an important role in terms of microstructural features that are present in Al-Si alloys. A number of conventional processes like casting and extrusion have been widely used to manufacture Al-Si alloys, which would normally involve multiple intermediate steps. While conventional processing is ideal for bulk production, one of the challenges of these processes is the limitation for fabricating parts of complex shapes. In addition, these processes are not designed and optimized for repair-oriented work on small to large components. Furthermore, these

processes are limited by their capabilities for rapid solidification, a key requirement for the control of the grain growth and reduction in size of the primary Si phase. During the heating and cooling process, grain growth may not be an issue if the initial particle size of the alloy is large, but submicron particles often exhibit uncontrolled and uneven growth during heating and solidification. The other factor that can aid grain growth is the slow diffusion of particles in the order of $10^{-3}$ and $10^{-4}$ m/s while static compaction methods like cold or hot isostatic pressing are employed as part of the conventional processing route.

When using traditional subtractive manufacturing technologies, the geometrical intricacy of many aluminium components can cause fabrication issues. But additive manufacturing (AM) processes lend themselves to many applications related to aerospace due to low volume and high value and a geometrical complexity of components can be manufactured [3]. AM for aluminium alloys has been explored less compared to other metals like titanium alloys and steel in various industries [4]. Aluminium alloys are used on a limited basis in AM processes due to a lack of alloys suitable for complex thermal cycles during the process [5]. Hydrogen-associated porosity is a disadvantage that must be considered when using aluminium alloys in additive processes. Direct energy deposition (DED) is one AM process that has huge advantages in fabricating aluminium alloys. In the DED process, a laser is used to melt a substrate and feedstock of material, i.e., a wire or powder is directed into the melt pool [6]. The DED technology allows for the addition of desired features to existing components as well as the production of near-net-shaped components on a substrate [3]. Few researchers have been trying Laser Direct Energy Deposition (LDED) with aluminium alloys, including Al-Si and Al-Si-Mg [7]. A great advantage of this technique is the greater flexibility and control it offers when it comes to managing heat input.

Several configurations of LDED exist depending on the type of powder feeding. With continuous coaxial powder feeding, as in the present experiments, several cones are coaxially mounted on the laser head to form gas and powder/gas channels that are roughly collinear with the laser beam. Other configurations exist, such as discontinuous coaxial feeding, where the coaxial nozzle contains several discrete channels, as well as lateral feeding, where the powder/carrier gas channel is formed with a distinct pipe and injector that are usually located on the side of the laser head. LDED is one of the most recent AM processes that is capable of producing high-density metallic alloys from particles of a wide range of size and shape, without requiring a mould of any kind. A great advantage of this technique is the greater flexibility and control it offers when it comes to managing heat input. The geometry of the part is usually based on a 3D CAD model that is sliced to obtain cross-sections of the object that will be built with LDED following various deposition strategies for the tool path. Such a layered structure creates an object that typically demonstrates strong metallurgical bonding between the layers [8]. During deposition, the highly localized heat input allows for a fast cooling rate, which is another reason why LDED has gained popularity. As opposed to subtractive manufacturing, LDED significantly reduces material wastage, which significantly reduces fabrication costs, even for complex structures. In addition, the optimization of process parameters such as laser power and scan speed can lead to an extremely high degree of adhesion between the layers and with the substrate. Overall, LDED has made its mark as a successful technique for printing wear and corrosion protection layers on metallic surfaces [9,10], for surface functionalization [11] and even for the repair of complex structures and parts.

The presented literature survey depicts that a good number of papers have reported on powder-bed-fusion-additive-manufactured AlSi10Mg alloy characteristics. In addition, limited studies are available on an LDED-processed AlSi alloy and its characterization. Therefore, the current work presents a study on LDED-processed eutectic Al-12Si thin walls. The aim of the work is to study the printability of thin walls using the LDED technique, and to subsequently examine the microstructural behaviour and analyse the mechanical properties subsequently compared with the cast alloy. The presence of different structures as well as porosities, and how they impact mechanical behaviour, will also be discussed.

## 2. Experimental Procedure

The process started with commercially available gas-atomised Al-12Si powders (procured from Praxair) with a particle size of 45–90 μm. A single feeder was used to transport powders towards an Aluminium 2024 rolled plate substrate for the deposition of a single-layer wall. The thickness of the substrate was 2 mm. The LDED process used an IREPA Laser Clad machine (Figure 1) and an IPG 2 kW continuous fibre laser, with a wavelength of 1070 nm and a laser spot diameter of 600 μm at the focus plane. The powder was carried by argon gas and transported to the laser beam through a coaxial nozzle.

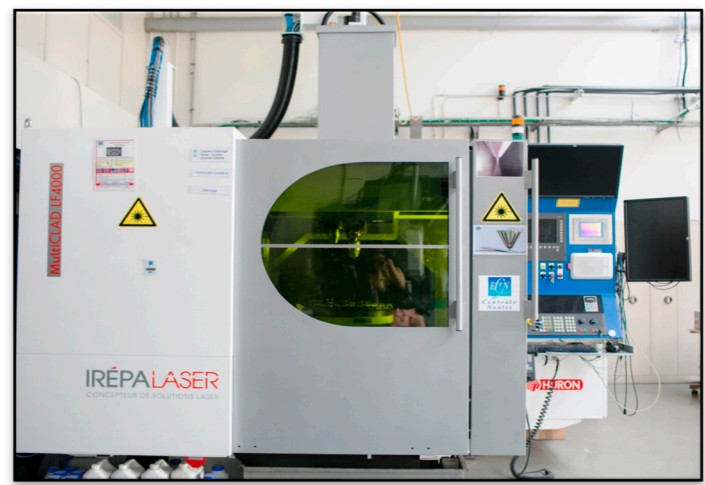

**Figure 1.** The IREPA Laser Direct Energy Deposition machine used in the study.

The LDED process can be used to manufacture either solid or hollow parts. A thin wall geometry and cubic samples were considered for fabrication in order to study the ability of the Al deposition, porosity, hardness, and microstructure of Al-Si deposits. A single wall with a length of 40 mm and a height of 20 mm was fabricated using the linear zigzag strategy with a perpendicular increment of 0.16 mm between layers. The selected process parameters for fabricating the wall are reported in Table 1.

Based on the literature and trail ad error, we chose 20 trails for initial experimentation to print the thin walls. Based on the results of 20 trials, the optimal conditions were chosen.

**Table 1.** LDED process parameters used for fabrication of wall.

| Laser Power (w) | Scan Speed (mm/min) | Powder Flow Rate (g/min) | Vertical Increment Δz (mm) | Number of Layers | VED (J/mm³) |
|---|---|---|---|---|---|
| 650–690 | 2000–2500 | 1.6–1.7 | 0.16 | 125 | 13.8–18.3 |

These process parameters can be combined into a measure of energy density using the relation VED = P/AV, where P is the laser power (W) and A = $\pi r^2$, where r is the laser spot radius at the focus plane and V is the scan speed (mm/sec). This allows for better comparing experimental conditions between various works. The term VED refers to volumetric energy density.

Fabricated specimens were subjected to a tensile test on an INSTRON 5565 machine (Norwood, MA, USA) with a load cell of 50 kN using an extensometer with a gage length of 10 mm with a strain rate of 0.001 mm/s. Tensile testing was performed to evaluate the strength of thin walls. Four specimens of each condition and the average values were taken for consideration.

A cross-section of the wall was first cut out for a microscopic analysis. Surface profile roughness measurements were first carried out using an Alicona Infinite Focus microscope [12,13]. The roughness was measured with an optical technique and non-contact measurement equipment with a white light interferometer, Alicona Infinite Focus

G6 Optical 3D Measurement (objective: 20× and a resolution of −50 nm). The measured area was a 1 × 1 mm width (measurement points: 3.2 million; lateral resolution: 2930 nm. The sample was then hot mounted, grinded, and finely polished with an OPS-U solution. For etching, a fresh batch of Keller's reagent was prepared and applied for 30 s on the surface of the sample to reveal the microstructure. The bubble formation during the etching process was carefully monitored, as the process needs to be stopped before the reaction becomes aggressive. Vickers micro-hardness measurements were carried out on the polished surface using a Fischerscope HM2000 automatic micro-hardness tester using a 100 g load with a dwell time of 20 s. The micro-hardness was measured along the build direction in order to understand the effect of microstructure on mechanical properties of the deposits. Three readings were taken for each section and then all values were averaged to find the hardness results of the surface.

The density of fabricated thin walls and cubic blocks was tested with the Archimedes principle. The density of the thin wall of 98.84% and blocks of 99.74% was achieved. Optical microscopy was carried out on both etched and unetched samples using a Carl Zeiss optical microscope and an Axio Vision image analyser. Quantitative porosity measurements were also carried out using Axio Vision from a selection of 1000 pores, which was later validated using ImageJ software. The powder morphology and fabricated samples' microstructure was examined using a JEOL 6000Plus scanning electron microscope (SEM) equipped with a secondary electron detector operating at 20 kV. The microstructural evolution was analysed in detail on the single-layer wall using a secondary electron detector. Powder particle size was analysed using a MIPAR Image Analysis. The powder flowability was tested using a Hall flow meter and the ASTM B213-13 standard.

## 3. Results and Discussion

Figure 2 shows the SEM image, elemental analysis, and particle size of gas-atomised Al-12Si alloy powder (Praxair) (45–90 μm) used in this study. Most particles are approximately spherical in shape with rare satellites. Particles that are within the 45–90 μm size range do not show signs of agglomeration, while some larger particles are seen as having satellites due to agglomeration during gas atomization. Despite the presence of satellites and a relatively small density compared to other types of alloys, the Al-12Si powder flows satisfactorily within the LDED system as it produces a fairly constant and uniform powder flux at the nozzle outlet, which is revealed with the regularity of the deposited thin walls.

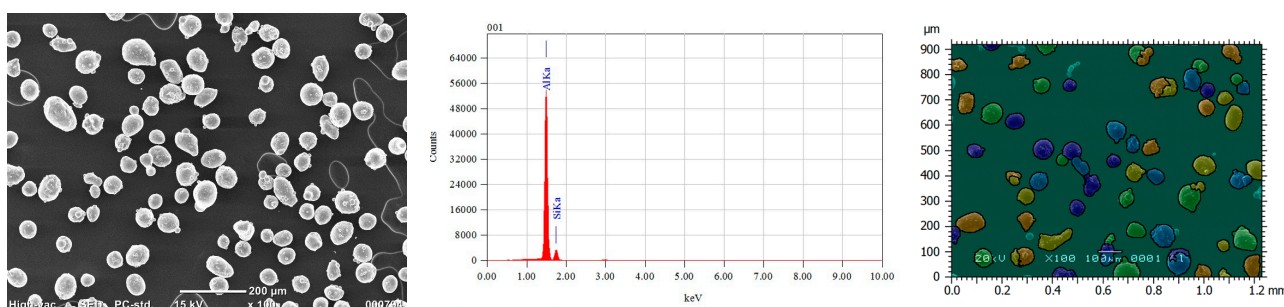

**Figure 2.** As-received Al-12Si powder morphology, EDAX analysis, and powder size measurement. (Different color represents the variation in particle size).

One of the important factors here is to understand the solidification techniques involved for producing these particles. Since the particles are gas-atomised, finer size will lead to faster solidification, essentially meaning that those particles will have more microstructures consisting of both stable and metastable phases. These different microstructural phases are generated due to the thermal and solute conditions that are changing during the process, and later during the LDED melting and re-melting stages. It can be said that the microstructural variation offers a clear indication of time-dependent changes [14,15]. The size range of the particles in this study is not a submicron scale (<1 μm), which means that

the eutectic composition of Al and Si will be consistent in not demonstrating massive grain growth. It is known that during heating, a submicron, especially nanoparticles, experience initial dynamic grain growth followed by normal growth, which often cannot be limited beyond a certain extent, without the presence of grain growth inhibitors [16]. Since the particle size is reasonably large in this study, massive grain growth within this temperature range is not expected. This also helps reduce the presence of metastable phases during the solidification of the atomization process or later during laser metal deposition [17].

The fabricated wall macrostructure is shown in Figure 3, which reveals multiple fusion bands between deposited layers, as well as a dendritic microstructure growing along the build direction. The fusion bands consist of both coarse and fine microstructures. During deposition, the microstructure in the partially re-melted zone (i.e., at the interface between two consecutive layers) evolves from coarse to fine due to the further accumulation of heat. When re-melting is insufficient, new grains are formed instead of continuing the growth of grains from previously deposited layers. The barrier against grain growth is also enhanced with the simultaneous cooling of layers as the heat does not dissipate through any die as it does in powder sintering or casting, but rather by using convection with the ambient atmosphere or by using conduction through the layers towards the substrate.

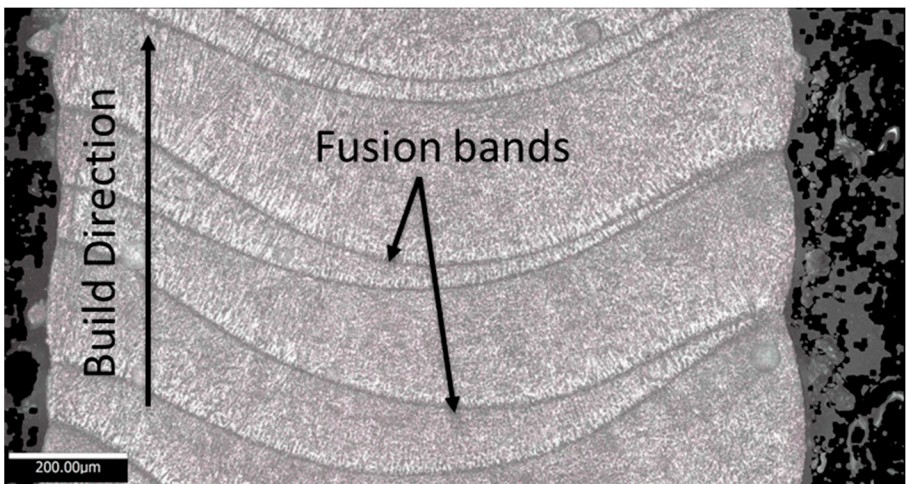

**Figure 3.** Macrostructure of fabricated wall at laser power of 800 w, 2100 mm/min scan speed, and powder flow rate of 1.4 g/min.

Figure 4 shows an optical image focusing on the dendritic structures of the material. It usually forms and expands in a tree-like structure during the solidification phase, right after the melting or re-melting of particles. Melting occurs when laser energy density imparts to powders to deposit them on the substrate or on top of a prior layer. While melting is performed on a previously formed layer, the surface of the solidified layer re-melts or partially melts since the temperature obtained is close to the eutectic temperature, even though rapid solidification simultaneously occurs between the layers. The size of the dendrites in the figure, as well as their homogeneity, dictate the mechanical behaviour of the samples. It can be clearly seen that the dendrites are homogenously distributed along the surface. The growth of the dendrites is not significantly high, which indicates that the material could potentially demonstrate reasonable ductile properties through an inverse relationship, a phenomenon explained by the classical theories of Spear and Gardner [18,19], while some argued that it could be a stepwise dependence on the scale of the dendritic structure. The Secondary Dendrite Arm Spacing (SDAS) is usually correlated with the cooling rate, which is expected to be on the order of $10^3$–$10^5$ K/s with LMD. The classic relationship $SDAS = A(CR)^{-n}$ links the cooling rate CR to the SDAS through material specific constants A and n, which are, respectively, taken as 40.7 and $-0.33$ for Al-12Si with SDAS expressed in μm (values based on Al-7Si) [20]. It is also thought that the parameter n is dependent on the cooling rate. For a SDAS of around 1.2 μm, the cooling rate is on

the order of $4.3 \times 10^4$ K/s, which is indeed within the expected range and the cooling rate is comparatively much higher than with casting (0.01–1 °C/s) or even welding processes (0.1–12 °C/s) [21,22]. It is seen in Figure 4 that there was a fine micro-cellular grain area below the fusion line, and the zone was composed of columnar structures [23], while a first coarse dendrite area arose above the band, and a second dendrite area arose at the farthest, away from the fusion line. It is affected by the constitutional supercooling in the solid liquid interface. The presence of frequent solidification fronts between the dendrites as well as between the melting and re-melting zones is indicative of the strength of the material [24]. The hardness of the samples was found to be around 95 HV, which is towards the upper end of strengths for eutectic Al-12Si [2]. The porosity content in the samples was found to be 2.6%. The banded microstructure that is often seen in such alloys is a combination of those dendritic structures, solidification lines, and re-melting zones.

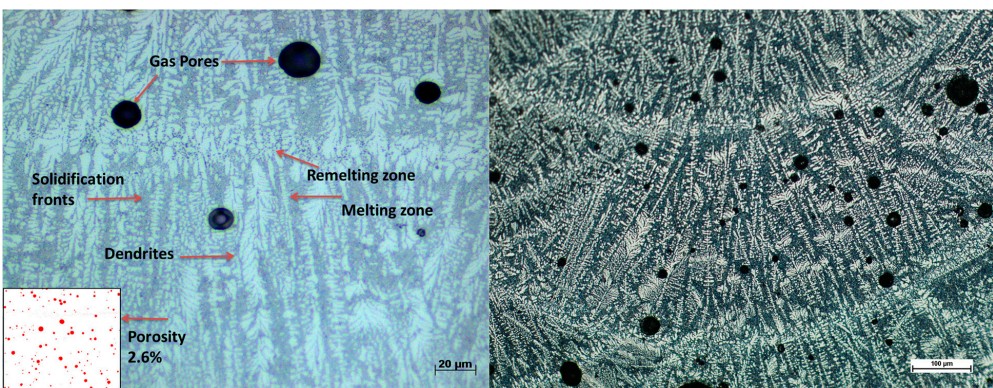

**Figure 4.** Optical micrograph showing presence and homogeneity of dendritic structures in Al-12Si (conditions: laser power of 750 w, 2100 mm/min scan speed, and powder flow rate of 1.4 g/min). The distribution of melting zones, solidification fronts, and dendrites confirms the strength of the material.

Figure 5a shows a 2D view of the surface of the sample, while Figure 5b demonstrates a 3D view of the section offering more information on the height and depth profile of the layers. What appeared on the 2D image as occasional pores are actually some small, unmelted particles, which are clearly seen around the 40 μm height range in the 3D image. The porosity content on the surface was quite low and the layers were well-bonded without the presence of any large voids. Pore formation comes from hydrogen being less soluble in solid than liquid aluminium, by a 20 to 1 ratio, so rapid-cooling hydrogen gas bubbles become trapped, as they do not have sufficient time to escape. Hydrogen can come from residual humidity that gets broken down into $H_2$ and $O_2$ upon interaction with the laser [5]. The entrapment of these gases often creates pores in the build part during cooling. Details on the surface profile are given in Table 2. It is observed that the average height within the selected area is around 10 μm for a maximum height of about 48 μm, which is significantly higher than the 2 μm usually required for parts' fitting or general surface profile roughness requirements. However, this is for the rough part, which can be further milled and polished depending on the requirements. Such deposits would thus require finishing operations to meet specifications in most cases. However, the above roughness is also significantly smaller than the average size of the particles, showing that reasonably good melting of the larger powder particles was achieved. Also, such roughness levels are very acceptable for a part fabricated with an LDED process [25].

Melt pool monitoring was performed on fabricated samples using varying laser powers. It was observed that lower laser powers resulted in smaller melt pool depths: 142 μm for (a) 800–600 w, 326 μm for (b) 1100–670 w, and 338 μm for (c) 1000–750 w. This indicates a substantial impact of laser power on melt pool geometry and depth. Within this parameter range, an escalation in laser power led to a proportional increase in melt pool depth, as depicted in Figure 6. This rise can be attributed to the augmented energy density. However, excessive laser power could potentially create a keyhole, giving rise to porosity

defects. Therefore, the identification of the optimal laser power and scan speed is crucial in controlling melt pool geometry to ensure the production of defect-free products. Notably, this observation suggests that the monitoring process correlates reasonably well with melt pool dimensions, but not necessarily with the detection of internal flaws.

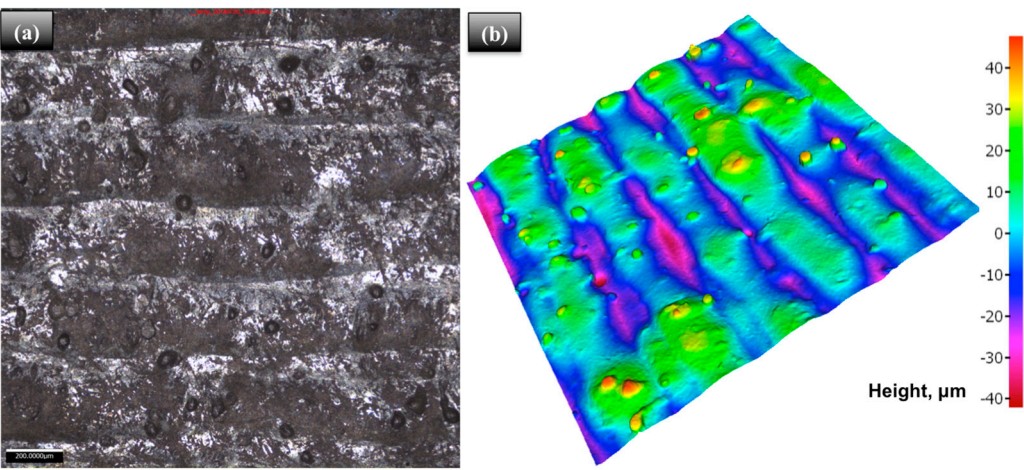

**Figure 5.** (**a**) 2D and (**b**) 3D view of laser-metal-deposited Al-12Si sample.

**Table 2.** List of surface profile roughness parameters of laser-metal-deposited Al-12Si.

| Name | Value | Unit | Description |
|------|-------|------|-------------|
| Sa | 10.6402 | μm | Average height within selected area |
| Sq | 13.3475 | μm | Root-Mean-Square height within selected area |
| Sp | 47.8771 | μm | Maximum peak height within selected area |
| Sv | 41.9621 | μm | Maximum valley depth within selected area |
| Sz | 89.8393 | μm | Maximum peak-to-peak height within selected area |
| S10z | 82.4880 | μm | Ten-point height within selected area |
| Ssk | 0.2691 | - | Skewness of selected area |
| Sku | 2.8706 | - | Kurtosis of selected area |
| Sdq | 0.5405 | - | Root-Mean-Square gradient |
| Sdr | 11.4868 | % | Developed interfacial area ratio |
| FlTt | 89.8393 | μm | Flatness using least squares reference plane |

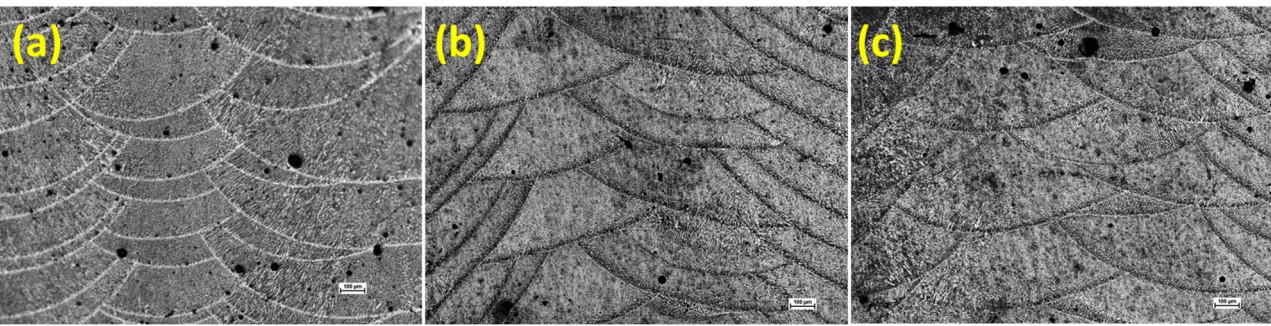

**Figure 6.** Melt pool profile and depth at different laser powers: (**a**) 800–600 w, (**b**) 1100–670 w, (**c**) 1000–750 w.

Figure 7a shows the SEM micrograph of the top-view microstructure of the LDED sample. The micrograph reveals the teardrop-shape molten pool microstructures caused by continuous laser beam interaction with the molten material reported in other studies [26]. In fact, the microstructure involves rounded molten pools overlying each other towards the path of laser deposition movement. The laser induces on/off periods for each successive layer. This will create each individual molten pool to solidify before the next layer deposits.

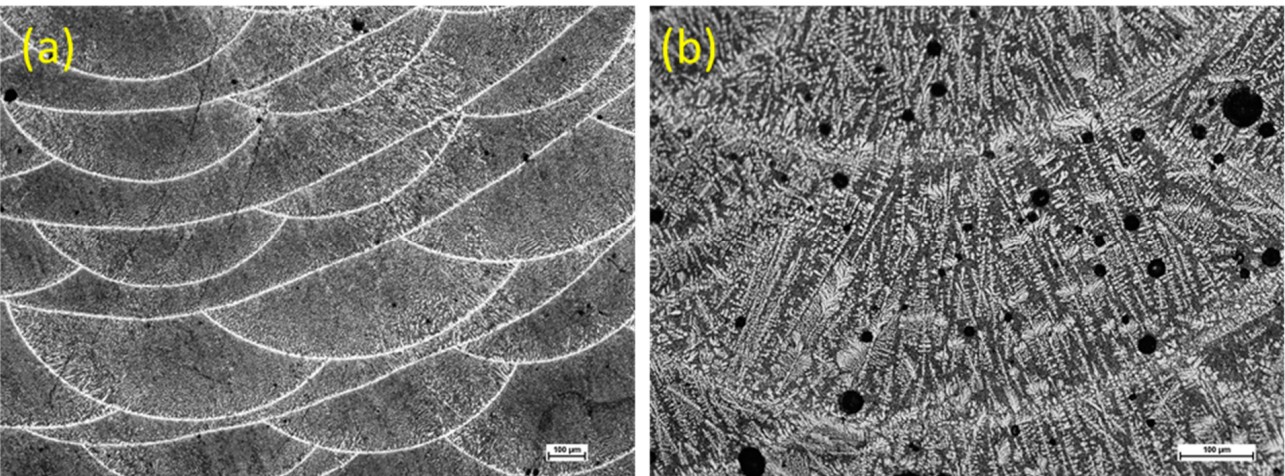

**Figure 7.** Microstructure of fabricated Al-12Si thin wall using LDED at laser power of 750 w, 2100 mm/min scan speed, and powder flow rate of 1.4 g/min; (**a**) outer top view and (**b**) cross-section at higher magnification.

Figure 7b demonstrates SEM cross-section micrographs towards the build direction. It particularly focuses on a grain structure of Al-12Si between the adjacent layers showing the dendritic microstructure oriented with an approximate 45° from the layers. The boundary of the microstructure shows a coarsened-zone approximately 2 μm width. The measured dendritic arm spacing is approximately 250 nm. This suggests that the coarsened area is causing change in solidification kinetics as opposed to a solid-state growth phenomenon of the layer below it. The microstructure of typical casted Al-12Si shows the eutectic phase, which consists of an Al matrix with flake-like Si particles throughout the microstructure. The microstructure of casts can be found in the literature [26]. During the normal casting, the solidification rate will be normally slow and the undercooling experienced will be small compared with the large undercooling experienced in LDED.

Figure 8a shows SEM images of the deposited Al-Si samples. The surface showed the presence of both a fine and coarse microstructure inside the deposit. It is noted that the microstructure is a typical dendritic structure. A magnified view illustrates the presence of a very fine eutectic structure. The grey cellular regions are composed of pure Al, and their boundaries contain fibrous Si particles (Figure 8b). Also, a uniform microstructure is observed along the build direction. The solidification of the alloy creates a continuous Al-Si eutectic phase—the eutectic region consisting of Al and Si precipitates. The eutectic point of the alloy is around 11.7 wt.% Si and eutectic temperature is 577 °C. It provides a natural composite, which gives good mechanical properties compared to an Al/Si powder mixture fabricated with sintering, casting, or additive manufacturing. Al-12Si is known to solidify dendritically, and the primary growth grains' direction may grow along the low index <100> perpendicular to the longitudinal surface of the deposit [25].

It is well known that the type and size of the solidification microstructure are largely determined using the cooling rate and the temperature gradient within the melt pool. In LDED, the highly focused heat input from the laser beam can create a large temperature difference within the melt pool as heat accumulates locally near the surface. Some of the heat escapes the melted zone, either towards the substrate with conduction or with convection through the sides of the wall, although the former is expected to be much larger. Indeed, the thermal conductivity of aluminium alloys is generally more than four times higher than that of stainless steel. Significant grain growth along the vertical or build direction is thus expected as previously deposited layers are reheated and even partially re-melted during the deposition of subsequent layers.

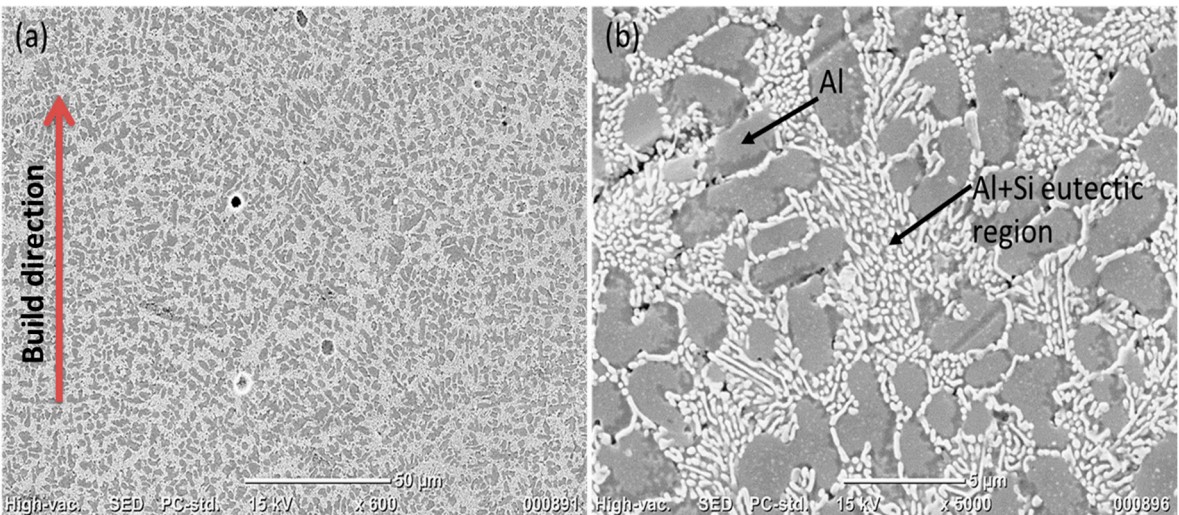

**Figure 8.** Microstructure of Al-12Si wall fabricated with LDED at laser power of 750 w, 2100 mm/min scan speed, and powder flow rate of 1.4 g/min. (**a**) Grain structure at low magnification (**b**) Fibrous Si particles distribution.

### 3.1. Mechanical Properties

Laser deposition parameters had to be optimized for the single wall as shown in Table 3. The determination of single wall microstructural and mechanical properties is critical because thin-wall-welded brackets in automobile applications can be replaced with a deposited bracket. Hence, very first layers were deposited at different laser powers (600–1100 W) and a lesser powder flow rate (1.4 g/min) to have a good bonding between the substrate and deposit. Consecutive layers were deposited using a reduced range of laser power. As the layer height increases, the non-uniformity of the layers starts to be more evident as shown earlier in Figure 3. This is primarily because of the drop in the heat dissipation rate, which causes more heat to be available in the wall at that time. The excess and rather uncontrolled heat accumulates at one end of the wall, which leads to uneven wall height and may also cause sintered powders to fall on the side. This can be due to an uneven melt pool and because of a high thermal conductivity of aluminium.

**Table 3.** Optimized laser deposition parameters for fabricating thin wall.

| Trials | Laser Power (W) | Scan Speed (mm/min) | Powder Flow Rate (g/min) Approximate | Hatch Spacing ΔZ (mm) | Ultimate Tensile Strength (MPa) | % Elongation |
|---|---|---|---|---|---|---|
| 1 | 1100–670 | 2100 | 1.4 | 0.18 | 96.96 | 9.1 |
| 2 | 1000–750 | 2100 | 1.4 | 0.18 | 155.5 | 10.9 |
| 3 | 800–600 | 2100 | 1.4 | 0.18 | 140 | 9.2 |
| 4 | | | Casted Al-12Si [1] | | 167 | 3.4 |

Thin-wall specimens were fabricated at different laser powers to have enough penetration depth for the initial layers and the power was reduced for subsequent layers. After successful trial and error by setting the process parameters, thin walls were fabricated at different laser powers to evaluate the strength as is shown in Figure 9. The tensile test was performed by extracting tensile coupons from fabricated thin walls. The Al-12Si alloys are more eutectic and are more printable but contain little Mg and are therefore not the solution for high-strength applications. Figure 10 shows the tensile stress–strain for different samples; a decreased tensile strength was observed for an increased laser power. The tensile properties of the additively manufactured Al12Si alloy have commendable mechanical properties over the casted specimens [1]. This superiority in mechanical strength is due to fine silicon particles uniformly distributed over each layer in LDED. It reveals that lower laser power does not have enough laser energy density to melt and penetrate into the substrate surface to have efficient bonding. These results indicate that the better strength in Al-fabricated parts can be attained by optimizing the LDED process parameters, and

practical implementation is feasible. The optimized laser deposition parameters are given in Table 3.

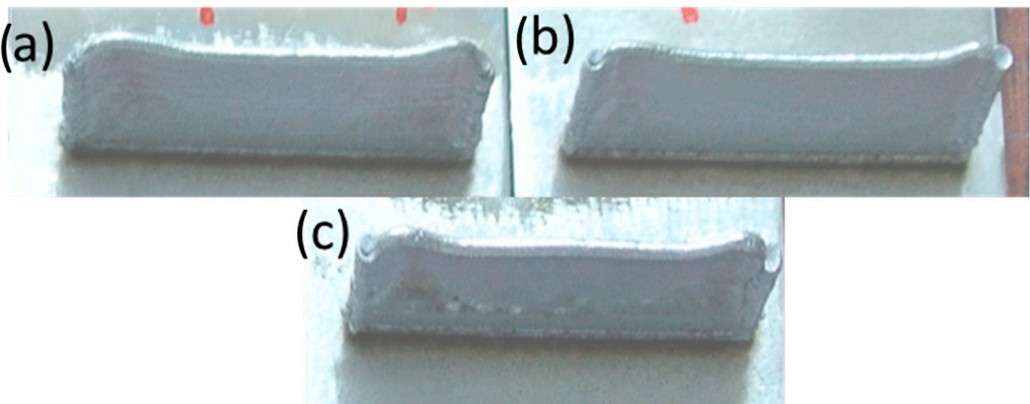

**Figure 9.** Fabricated samples to evaluate the tensile strength of varying laser powers: (**a**) 1100–670 w, (**b**) 1000–750 w, (**c**) 800–600 w.

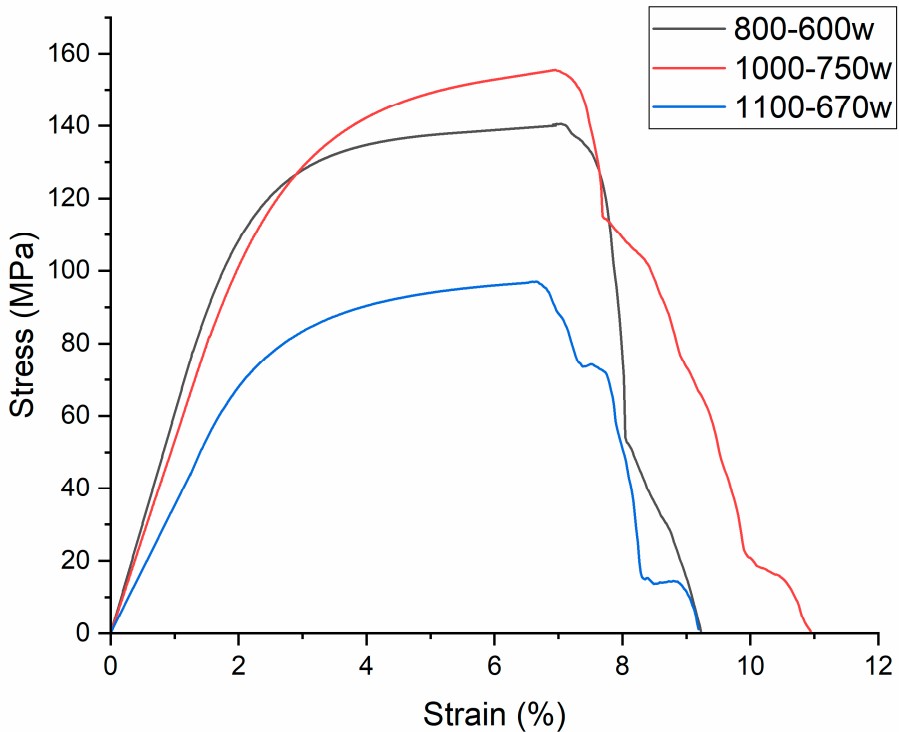

**Figure 10.** Stress versus strain of the fabricated thin wall at different laser powers.

### 3.2. Micro-Hardness

The hardness of fabricated samples was measured at the different cross-sectioned surfaces with a load of 100 g with a dwell time of 10 s using Fischerscope HM2000 as is shown in Figure 11. The measured values are among 64–95 HV, and the average value of common-aluminium–12%-silicon fabricated with a conventional casted alloy reaches only 65–80 HV [26]. The hardness was measured through the build direction (Z direction); there are some variations in hardness values that were observed. This may be due to the fact that the indentation force acts in the melted layer, re-melted layer, or fusion line, or on the porous zone. The results show that laser-metal-deposited Al-12Si has better hardness due to rapid solidification inducing a refined grain size, which dominates the higher level of hardness. Referring to the microstructures depicted in Figure 8, it is expected that the early

deposited layers, experiencing greater heat dissipation, will develop a fine grain structure, consequently exhibiting higher hardness. Conversely, as the layer height increases, the reduction in heat dissipation is likely to result in the formation of a coarse grain structure, leading to diminished hardness properties, aligning with the principles of the classical Hall–Pitch theory [27,28]. The published results [29] of an Al alloy fabricated with a selective laser melting (SLM) process involve 133 HV, which is near to LDED-fabricated alloys because the solidification takes place in SLM and LDED is almost the same. The hardness of Al-Si alloys greatly depends on not only the grain size but also the size and distribution of the Si phase throughout the matrix. These results show evidence that the LDED could promote and improve mechanical properties compared to casts because of refined grains due to rapid solidification.

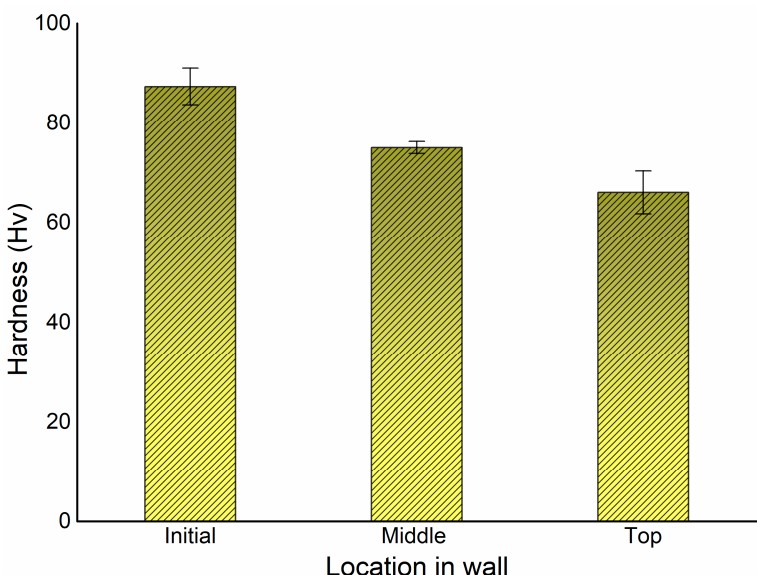

**Figure 11.** Micro-hardness of fabricated thin wall at different zones.

### 3.3. Phase Analysis

Figure 12 shows the XRD pattern of the Al-12Si thin wall sample prepared via laser DED. As expected, the XRD pattern shows the peaks of Al and Si. The formation of a supersaturated solid solution of Si in Al, caused by the rapid cooling rate, resulted in a reduction in the intensity of Si peaks. This is indicative of the cellular morphology of the microstructure, as shown in Figure 8a. The Si peaks are broader, which results in the reduced size of the Si phase in fabricated samples. The results are similar to those already reported by other researchers [18–21]. However, no phase formation is observed for fabricated samples. On the other hand, the PBF, heat-treated, and cast sample will also have the same peaks, which shows good agreement with the amount of Si solubility in Al.

### 3.4. Residual Stress Analysis

XRD stress measurements were performed with proto iXRD (MGR40P) using the $K\alpha$ emission line of a Cr filament ($\lambda = 2.291$ A$°$) as incident radiation and focusing on the (222) reflection of the face-centred cubic (FCC) Al-rich phase. XRD was performed on the build direction of a thin wall at different locations. The residual stress was measured using the distance between the crystallographic planes, i.e., d-spacing, as a strain gauge. The detector was often moved in relation to the sample and source of the X-ray, collecting multiple reflecting angles, and the sample's XRD pattern is shown as the intensity (of the reflected beam) versus twice the reflection angle. Change in inter-atomic distance may be attributed to elastic stresses and, hence, to residual stresses. Figure 13 is showing the plot between measured inter-planar d-spacing values and $\sin^2\psi$ at particular locations. Residual stresses (i.e., $\sigma_\phi$) were measured using the slope of a plot of measured $d_{\phi\psi}$ versus

$\sin^2 \psi$, which gives $\left(\frac{1+\nu}{E}\right)\sigma_\phi$, as given in Equation (1) [30]. Tensile residual stresses were found on the surface of the Al-12Si sample and the average magnitude of residual stress is $+52.3 \pm 2.7$ MPa. The exterior surface of components cools fast during the LDED process, causing contraction, which causes plastic deformation inside the core, resulting in residual stresses. As a result of cyclic heating and cooling, the components may become locally distorted, where the tensile residual stresses created exceed the material's yield strength.

$$d_{\phi\psi} = (\frac{1+\nu}{E}) \sin^2\psi - (\frac{\nu}{E})(\sigma_{11} + \sigma_{22}) \tag{1}$$

where

d = the inter-atomic distance relating to a specific (hkl) plane.

$d_{\phi\psi}$ = the stressed inter-atomic distance for the same (hkl) planes, while the incident beam is rotated using $\phi$ and tilted using $\psi$

$\nu$, E = Poisson's ratio and Young's modulus of material, respectively.

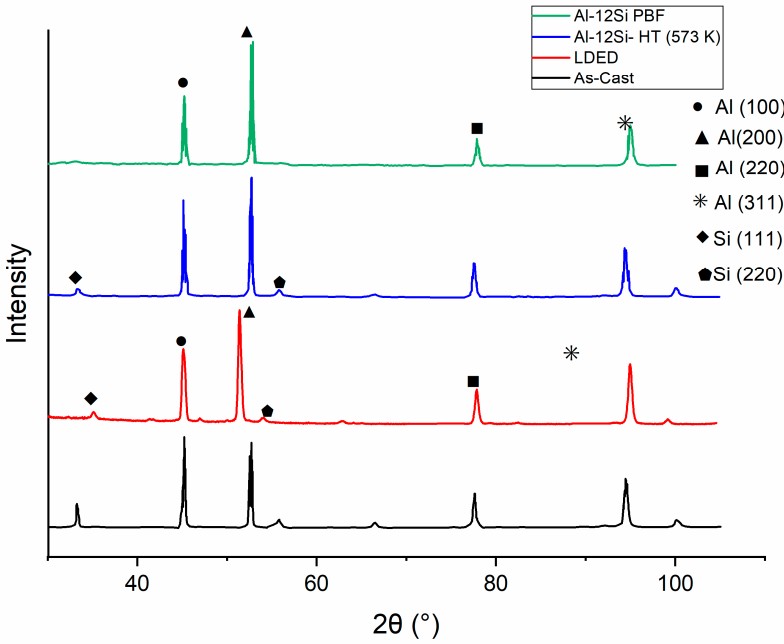

**Figure 12.** XRD patterns of LDED, heat-treated, PBF, and as-cast Al-12Si samples.

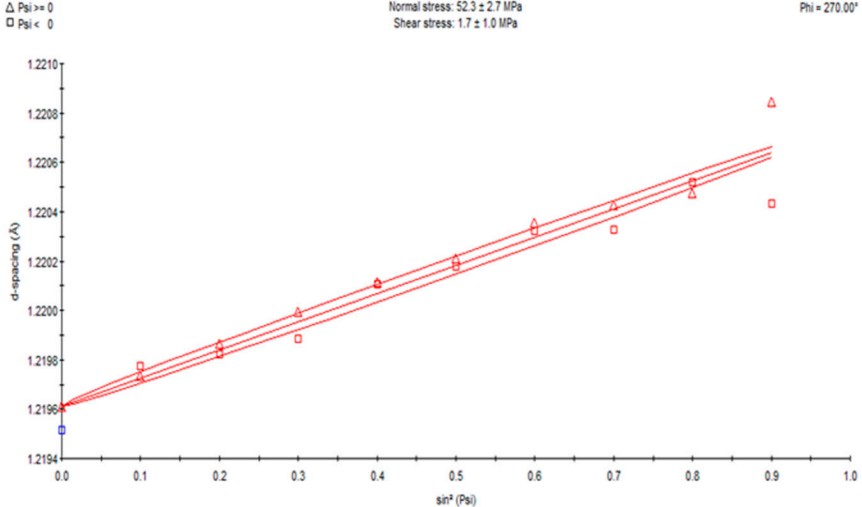

**Figure 13.** A plot of the inter-atomic distance (d) versus the $\sin^2\psi$.

## 4. Challenges and Opportunities

The primary challenges encountered in the additive manufacturing of aluminium (Al) stem from its reflectivity and low absorption properties, which hinder the effective laser melting of Al powder. Additionally, Al powders are susceptible to oxidation, impacting their mechanical properties. The evaporation of alloying elements with low boiling points like zinc and magnesium during laser melting can result in defects like porosity, cracks, a lack of fusion, voids, unsatisfactory grain growth, and shrinkage. The utilization of Al-Si and Al-Si-Mg alloys in additive manufacturing is facilitated by their relative ease of laser melting, rendering them suitable for processes like Laser Powder Bed Fusion (LPBF) and Directed Energy Deposition (DED). Their near eutectic composition attributes a small solidification range, distinct from high-strength Al alloys such as the 7075 series. Incorporating Al into DED processes introduces challenges due to its high reflectivity and low laser absorption, which can lead to defects. To achieve a better density, a minimum energy density of 42 J/mm$^2$ and layer thickness of 50 μm are necessary. Oxidation poses challenges as an oxide film forms on the Al powder. The small particle size and lightweight nature of the powder enhance its susceptibility to oxidation. The oxidation's higher melting point compared to the powder creates an uneven melt pool where the Al powder is fully melted while parts of the oxidation layer remain solid. This trapped oxide can induce porosity in the final part. A combination of high laser power and low scan speed can lead to a large melt pool, resulting in a balling effect that affects the powder and introduces defects. The presence of defects like porosity, cracks, and unsatisfactory grain growth is related to the microstructure, while oxidation involves a chemical reaction and shrinkage constitutes a geometric change. The additive manufacturing of Al holds significant interest across industries like the automotive and aerospace industries due to its lightweight properties and mechanical characteristics, which can rival or surpass those of cast Al products. Notably, studies have focused on AlSi10Mg and Al12Si alloys owing to their weldability. Exploring new alloy development presents an opportunity in Al AM. However, challenges remain, particularly concerning laser–powder interaction and process parameter adjustment for novel alloys. Paying heed to oxidation and shrinkage effects is pivotal in advancing Al AM processes.

## 5. Sustainability of LDED

Metal additive manufacturing is regarded as a more sustainable technology due to its capacity to create intricate components in a single step, reducing the necessity for machining and minimizing the use of environmentally harmful materials like acidic cutting fluids. Direct Energy Deposition (DED) stands out by swiftly and economically constructing intricate parts while producing less waste. Its applications span from high-value component repair to crafting prototypes and functionally graded materials. Industries such as aerospace, defence, automotive, and biomedical fields benefit from its capabilities.

Particularly, conventional manufacturing processes face challenges when fabricating thin walls or introducing additional features like fins to existing components. Yet, Laser-Based Directed Energy Deposition (LDED) emerges as the ideal method for adding fins or conducting repairs, facilitated by this work's optimal process conditions.

Furthermore, the wastage of material during thin-wall fabrication is almost negligible with LDED, a notable contrast to methods like Laser Powder Bed Fusion (LPBF), which lack the reparability LDED provides. The use of metal powder in LDED is already well established within industries due to its advantages in design flexibility, high performance, and the ability to produce intricate parts. This technology now extends its benefits to the environmental realm, enhancing its overall performance.

## 6. Conclusions

A thin wall made of Al-12Si was successfully fabricated using an LDED process with minimum defects. It was observed that the overall geometry of the thin wall is satisfactory

with respect to the design model. The examination of fabricated samples reveals the following conclusions:

- The surface profile roughness is relatively high compared to some conventional processes but it is nonetheless near the best levels that can be obtained for the LDED process.
- The microstructure presents two phases, namely pure Al and eutectic Al/Si that are uniformly distributed throughout the sample. The presence of a high-concentration eutectic phase mixture plays a major role in the enhancement of mechanical properties with the restriction of the dislocation moments.
- The solidification of the fused particles and melt pool formation was the main factor that lies behind the surface integrity of the fabricated components.
- Some unmelted/partial-melted particles caused the reduction in strength due to crack initiation and propagation.
- The tensile strength and ductility of condition 2 were found to be better compared to the other two samples due to the presence of less porosity and the fine grain structure.
- The hardness values are quite impressive, and are consistent across the entire sample. Some porosity is present but is comparable to values found in the literature for aluminium alloy parts made with AM processes.
- The phase formation and stress measurements reveal the tensile residual stress as perpendicular to the build direction.

**Author Contributions:** R.R., M.M. and S.T.: Conceptualization, Methodology, Software, and Formal analysis; R.R. and M.M.: Validation; R.A.S.: Investigation; R.R., M.M. and S.T.: Data curation, Writing—original draft preparation, Writing—review and editing, Visualization; J.-Y.H. and J.S.Q.: Supervision, Project administration, funding acquisition. All authors have read and agreed to the published version of the manuscript.

**Funding:** This research received no external funding.

**Institutional Review Board Statement:** Not applicable.

**Informed Consent Statement:** Not applicable.

**Data Availability Statement:** Data are contained within the article.

**Acknowledgments:** The authors acknowledge the expertise, equipment, and support provided by Ecole Centrale Nantes, France and Microscopy Australia and the Australian National Fabrication Facility (ANFF) at the South Australian nodes of the MA and ANFF under the National Collaborative Research Infrastructure Strategy.

**Conflicts of Interest:** The authors declare no conflict of interest.

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
