# Peer review of "Characterization of Al-12Si Thin-Wall Properties Fabricated with Laser Direct Energy Deposition"

_sustainability, doi:10.3390/su151712806_

Round 1

Reviewer 1 Report

This paper presents the microstructural and mechanical characteristics of Laser Direct Energy Deposited Al-12Si alloy. The paper is well written, and the opted material has high scientific soundness. This paper can be accepted after moderate revision. To improve the quality of the manuscript the following review comments are amended below.

1) In the introduction the kernel aim and objective to define the prime novelty of the study is missing.

2) In the experimental section it is good to mention how the optimal process parameters were chosen.

3) To promote better understanding to the readers it is good to present the melt pool depths and microstructural morphologies of 1100-670, 800-600 and 1000-750W.

4) It would be nice if all the figure captions are labelled according to the power specification.

5) In line 277, “grains grow along the low index <100> direction”, if you have any EBSD data to substantiate your claim then please include. It is not possible to always have a grain growth in one particular texture. In accordance with process parameters the texture direction could be different.

6) As aluminium alloys are known for high reflectivity, it would be very much interesting to the readers if you could include the any reflectivity issues encountered during this study.

7) Mechanical properties are poorly correlated. It is necessary to correlate the mechanical properties with respect to the grain size, pore density, grain orientation, precipitation behaviour and so on. It would be nice to correlate the mechanical characteristics with classical theories such as Hall Petch empirical phenomenon, Orowan strengthening effect and Zener pinning mechanisms. I encourage you to have a look at the following articles for better understanding.

https://doi.org/10.1016/j.jallcom.2023.169852

https://doi.org/10.1007/s12666-022-02756-6

8) Residual stress characterization seems to look unclear. Which location the residual stress distribution was calculated either along the build direction or the perpendicular to the build direction? Please double check the residual stress results.  There is no tensile to compressive transition across the different zones. It appears to be flat which is not convincing.

9) Fig.11. authors displayed only one XRD data and talking about peak broadening which is unacceptable. It is good to compare the XRD with wrought alloy or XRD data of other two samples to benchmark the results.

10) Few grammatical and typographic errors are there throughout the paper. For example in conclusion section: “due to crack imitation”.

Reviewer 2 Report

Dear Authors, the manuscript ‘Characterization of Al-12Si thin wall properties fabricated by Laser Direct Energy Deposition’, Manuscript ID: sustainability-2563286, have some weakness that must be revised properly.

Below are listed some of the most crucial comments:

1.      In the Abstract section, the motivation of studies mentioned in lines 15-21 is unclear. The Authors are trying to introduce the potential of the analysis, but its origin and future perspective are not listed.

2.      Further to the Abstract section, the motivation mentioned in the last sentence, The results can be used as a base guide for the further fabrication of aerospace component design with high structural integrity., lines 27-28, is not convincing. The Authors should add some more detailed advantages of the studies performed.

3.      For the ‘Introduction’ section the motivation, presented in lines 89-92, does not derive from the previous gaps of the section. The Authors are not trying to emphasize the lack in the current state of knowledge, where the presented studies should be located, but separate it one from another.

4.      As for the previous comment, the critical review of the literature does not allow Them to put forward any motivation. Usually, the strong and weak of the previous studies must be evaluated and then the studies proposal can be raised. Please try to highlight the motivation against lack in the already published papers, if exist.

5.      From section no.2, Experimental Procedure, many values selection are not justified and, respectively, looks like chosen arbitrarily. Authors should indicate (or referee) why used values are confidential to the study proposed, e.g. The thickness of the substrate, laser wavelength or spot diameter, etc.

6.      In the same section, some information is not detailed enough. For example, Alicona surface roughness measurements were not described appropriately, including instrument measuring parameters, measurement uncertainty, noise or other errors. Please try to referee to the mentioned issues more comprehensively, like in:

(1)   https://doi.org/10.1088/2051-672X/3/3/035004

(2)   https://doi.org/10.3390/coatings12060726

(3)   https://doi.org/10.1016/j.cirp.2014.03.086

7.      In the section no.3, Results and Discussions, there is no discussion that, respectively, there is no critical response to the results analysed. In detail, the Authors did not provide any limitations of their work.

8.      According to the previous comment, the advantages and disadvantages of the procedure proposed and the results studied are not accomplished. The Authors must emphasize the novelty according to the previous studies, those from Authors or other scholars.

9.      Looking for the description of Table 2, what are the ‘surface profile parameters’? As far as I am concerned we have surface roughness or profile roughness. The surface roughness, mandatory represented by the Sa (arithmetic mean height) is not the same as the Ra (arithmetic mean height of the profile). Extracting some of the profiles (it was also not clearly explained how the profiles were selected) is not referred to the surface roughness but the roughness of the profile received from the surface, which roughness can be strongly differentiated from the profile.

10.  For the subsection ‘Residual stress analysis’ located in section 3, the equations should be referenced, if not newly proposed, especially to the primary sources. The Authors should indicate their novelty proposal against results already published.

11.  The main proposal of the paper must be more clearly emphasized in the ‘Conclusion’ section. Firstly, the Authors should divide this section into separated and numbered gaps. Then, secondly, one of the gaps must be linked to the main novelty. Finally, some detailed information must be added as well.

Moreover, some additional issues must be listed, below:

12.  In the Abstract section, the ‘Additive manufacturing’, line 11, should be abbreviated that was presented in short form, ‘AM’, in the next sentence.

13.  Full DOI links should be added for all of the references, if exist.

14.  In section 3, Results and Discussions, each of the subsections should be numbered as well.

From the above, the reviewed manuscript must be improved significantly before any further processing of the Sustainability journal, if allowed by the Editor.

Round 2

Reviewer 1 Report

The essential modifications were made, and the authors' response to the reviewer's question is more convincing. 

Reviewer 2 Report

The manuscript can be accepted in the current, revised form.